# Measurements of Cloud Radiative Effect across the Southern Ocean (43° S–79° S, 63° E–158° W)

**Haoran Wang** [1] , **Andrew R. Klekociuk** [2,3,*], **W. John R. French** [2,3], **Simon P. Alexander** [2,3] and **Tom A. Warner** [4]

1   Institute of Marine and Antarctic Studies, University of Tasmania, Hobart, TAS 7001, Australia; haoranw@utas.edu.au
2   Department of Agriculture, Water and the Environment, Australian Antarctic Division, Antarctic Climate Program, Kingston, TAS 7050, Australia; john.french@awe.gov.au (W.J.R.F.); simon.alexander@awe.gov.au (S.P.A.)
3   Australian Antarctic Program Partnership, Institute of Marine and Antarctic Studies, Hobart, TAS 7001, Australia
4   ZT Research, Rapid City, SD 57702, USA; tom.alan.warner@gmail.com
*   Correspondence: andrew.klekociuk@awe.gov.au

**Abstract:** The surface radiation environment over the Southern Ocean within the region bound by 42.8° S to 78.7° S and 62.6° E to 157.7° W is summarised for three austral summers. This is done using ship-based measurements with the combination of downwelling radiation sensors and a cloud imager. We focus on characterising the cloud radiative effect (CRE) under a variety of conditions, comparing observations in the open ocean with those in the sea ice zone. For comparison with our observed data, we obtained surface data from the European Centre for Medium-Range Weather Forecasts fifth reanalysis (ERA5). We found that the daily average cloud fraction was slightly lower in ERA5 compared with the observations (0.71 and 0.75, respectively). ERA5 also showed positive biases in the shortwave radiation effect and a negative bias in the longwave radiation effect. The observed mean surface CRE of $-164 \pm 100$ Wm$^{-2}$ was more negative than the mean surface CRE for ERA5 of $-101$ W m$^{-2}$.

**Keywords:** Southern Ocean; Ross Sea; cloud; shortwave radiation; longwave radiation; cloud radiative effect

## 1. Introduction

The Antarctic and Southern Ocean regions play an important role in the Earth's climate system, contributing to the systems that regulate heat flow around the planet. Under global climate change, the polar regions are expected to respond most rapidly to changes in energy balance due to large contrasts in surface albedo, which influences how energy is absorbed near the surface and is transmitted into space [1]. A key factor in the overall albedo of the Southern Ocean is the extensive and persistent cloud cover over the region [2]. On a global scale, clouds with relatively high albedo cover more than half of Earth and influence the radiative budget mainly in two ways: (1) reducing the incoming solar radiation by reflection back to space and, (2) increasing thermal radiation by absorbing infrared radiation emitted by the Earth and re-emitting it in all directions. With these competing effects, clouds can introduce large uncertainties in simulations of climate change [3–5]. The total effect that clouds have on Earth's radiative budget, calculated from the difference of incoming solar radiation and thermal radiation between average conditions and cloud-free conditions, is called the cloud radiative effect (CRE) and is defined as [6,7]:

$$CRE(\theta) = (1 - \alpha) \times (S(\theta) - S_0(\theta)) + \varepsilon \times (L(\theta) - L_0(\theta)) \tag{1}$$

where $\alpha$ is the surface short-wave (SW) albedo, $S(\theta)$ and $S_0(\theta)$ are the downwelling SW irradiance incident upon Earth's surface in the presence and absence of clouds, respectively, $\theta$ is the solar zenith angle, $\varepsilon$ is the long-wave (LW) emissivity of the surface, and $L(\theta)$ and $L_0(\theta)$ are the downwelling LW irradiance, in the presence and absence of clouds, respectively. A positive (negative) CRE indicates that the sea surface is cooled (warmed) by the clouds. $S_0(\theta)$ contains contributions by both direct and indirect (diffuse) solar radiation, which are additionally influenced by atmospheric transmission and specular reflection at large solar zenith angles, while $L_0(\theta)$ is primarily dependent on the effective temperature of the atmosphere. Equation (1) is the sum of short-wave cloud radiative effect (SCE) and long-wave cloud radiative effect (LCE [4]):

$$SCE(\theta) = (1 - \alpha) \times (S(\theta) - S_0(\theta)) \tag{2}$$

$$LCE(\theta) = \varepsilon \times (L(\theta) - L_0(\theta)) \tag{3}$$

Cloud properties (droplets types and optical depths) and cloud fraction (amount of sky obscured by cloud) play a dominant role in the Earth's surface radiation budget by influencing the net CRE. Previous studies show that with an increased cloud fraction, SCE generally decreases but LCE increases [3,8]. Cloud types also influence CRE as the composition of water droplets and ice crystals for different types of clouds vary, leading to different cloud optical depths which are directly related to the fraction of reflecting SW. Clouds near the Earth's surface, containing more water droplets which are relatively warm and bright, tend to have a cooling effect on the surface [9,10]. These low clouds have a strong influence on reflecting the incoming SW back to space, however, they have a smaller impact on LW as they have a similar temperature to the Earth's surface. Higher clouds, containing more ice crystals, are relatively colder and thinner and are often transparent to SW but can effectively absorb and emit LW, and therefore, these clouds tend to have a relative warming effect on the surface. Actual sky conditions are complex due to the existence of various types of clouds of different phases present at different altitudes, which results in different net CRE.

The albedo of the Earth's surface is another important factor that can strongly influence net CRE. Over the Southern Ocean, seasonal ice cover is a significant feature and plays an important role in the Southern Ocean radiation budget. Sea ice insulates the ocean from the atmosphere, reducing heat transport from the relatively warm seawater to the cold Antarctic atmosphere, while at the edge of sea ice cover heat transport is relatively enhanced [11]. According to Brandt et al. [12], the albedo of the sea surface can rise from 0.07 in open water to 0.49 with sea ice, and the albedo can further increase to 0.87 with thick snow cover, therefore, sea ice cover can effectively reduce the absorbed solar radiation by increasing the fraction of reflected solar radiation. Previous studies show that sea ice can significantly increase SCE by multiple scattering of SW between the surface and clouds [13].

Determining the future climate of high southern latitudes relies on state-of-the-art climate models. In this region, however, especially in summer seasons, many climate models show significant biases in sea surface temperature and energy budget that are related to the cloud fraction and cloud properties [14,15]. According to Bourassa et al. [16], the distribution of stations measuring radiation and data from surfaced-based remote sensing of clouds is sparse in southern high latitudes. This sparsity causes issues in evaluating climate models due to the inherent spatial and temporal variability of cloud cover. Satellite-based remote sensing data have difficulties observing low-level clouds, which are found ubiquitously over the Southern Ocean [17]. Super-cooled liquid water (SLW) clouds are particularly common over the Southern Ocean, compared with other oceanic regions around the world [18–21]. This higher occurrence rate of SLW clouds is likely due to the relatively lower aerosol loading found over the Southern Ocean [22]. Local CRE is amplified significantly due to the presence of SLW clouds [23,24] and requires careful treatment in climate simulations. Indeed, radiation bias in leading climate models at high southern latitudes is implicated as being due to inadequate representation of

SLW in the cold sectors of extra-tropical cyclones [25,26]. In the Southern Ocean region, where there are abundant low-level liquid clouds, satellite-based remote sensing data have difficulty in resolving the cloud thermodynamic phase [23,24,27]. Ship-based observations in this region are important for quantifying cloud and radiation properties and for evaluating climate and forecasting models [28–30].

Our study utilises day-time observations from three Antarctic summer seasons over a wide range of latitudes and sea conditions to determine the main influences on the surface CRE over the study region. We expand on the results of Klekociuk et al. [31], which only considered a small subset of the data that we investigate here. The local humidity in the cloud formation region has a bearing on CRE, at least in the Arctic as discussed by Cox et al. [32], while winds influence the distribution of cloud condensation nuclei by lofting sea salt and biogenic particles [33]. The spatial distribution of winds aloft varies over the Southern Ocean due to the complicated wind environment associated with the passage of synoptic weather systems, which makes it difficult to directly evaluate the influence of winds on CRE; a similar problem occurs when evaluating humidity. Additionally, large scale climate modes, such as the Southern Annular Mode which varies the location of the westerly wind belt over the Southern Ocean on timescales of weeks to months, can have specific effects that complicate statistical interpretation.

The organisation of this paper is as follows: firstly, we describe our ship-based measurements and associated analysis methods; secondly, we develop simple but suitable linear models for calculating the clear-sky SW and LW to obtain net CRE for each observation; then, we provide an overview of cloud fraction over the survey region and investigate the relationship between the presence of clouds and observed CRE, as well as comparing with previous studies to investigate cloud types; we then investigate the influence of sea ice on recorded SW by comparing the raw cloud transmittance with the open ocean, and the influence on net CRE. Finally, we characterise the cloud and radiation environment during three summer seasons and compare ship-based measurements with meteorological reanalysis data and previous studies, before summarising our conclusions.

## 2. Experiments

### 2.1. Overview of Voyage Data

To address the paucity of surface-based cloud and radiation data collected over the Southern Ocean, incoming SW and LW broadband radiometers and an all-sky cloud imager were deployed aboard Research and Supply Vessel (RSV) Aurora Australis as the icebreaking ship made regular transits from Hobart, Australia, to the Australian Antarctic research stations of Casey, Davis, and Mawson in coastal East Antarctica, and Macquarie Island in the Southern Ocean. These instruments provide measurements across the Southern Ocean between 43° S and 69° S during the austral summer period from November 2015 to February 2016 (Figure 1a; this figure also shows the return leg of the final voyage in March 2016 during which measurements were not made) and October 2018 until March 2019 (Figure 1b). Measurements were also obtained from a voyage of Motor Ship (MS) The World to the Ross Sea region (43° S to 79° S) in January and February 2017 (Figure 1c). Navigation and local meteorological data were also recorded for all voyages.

### 2.2. Cloud Imagers

An all-sky camera on RSV Aurora Australis was used to obtain colour images at 1-min intervals for analysis of cloud properties during the 2015 to 2016 and 2018 to 2019 seasons. The camera is based on a colour charge-coupled device (CCD) sensor with a three-element 1.24 mm F2.8 lens which can provide a 190° hemispherical "fisheye" field of view to identify cloud distribution and is weather-protected by a heated glass dome. Figure 2a shows the camera arrangement for the 2018 to 2019 season; the arrangement for the 2015 to 2016 season is shown in Figure 2 of Klekociuk et al. [31].

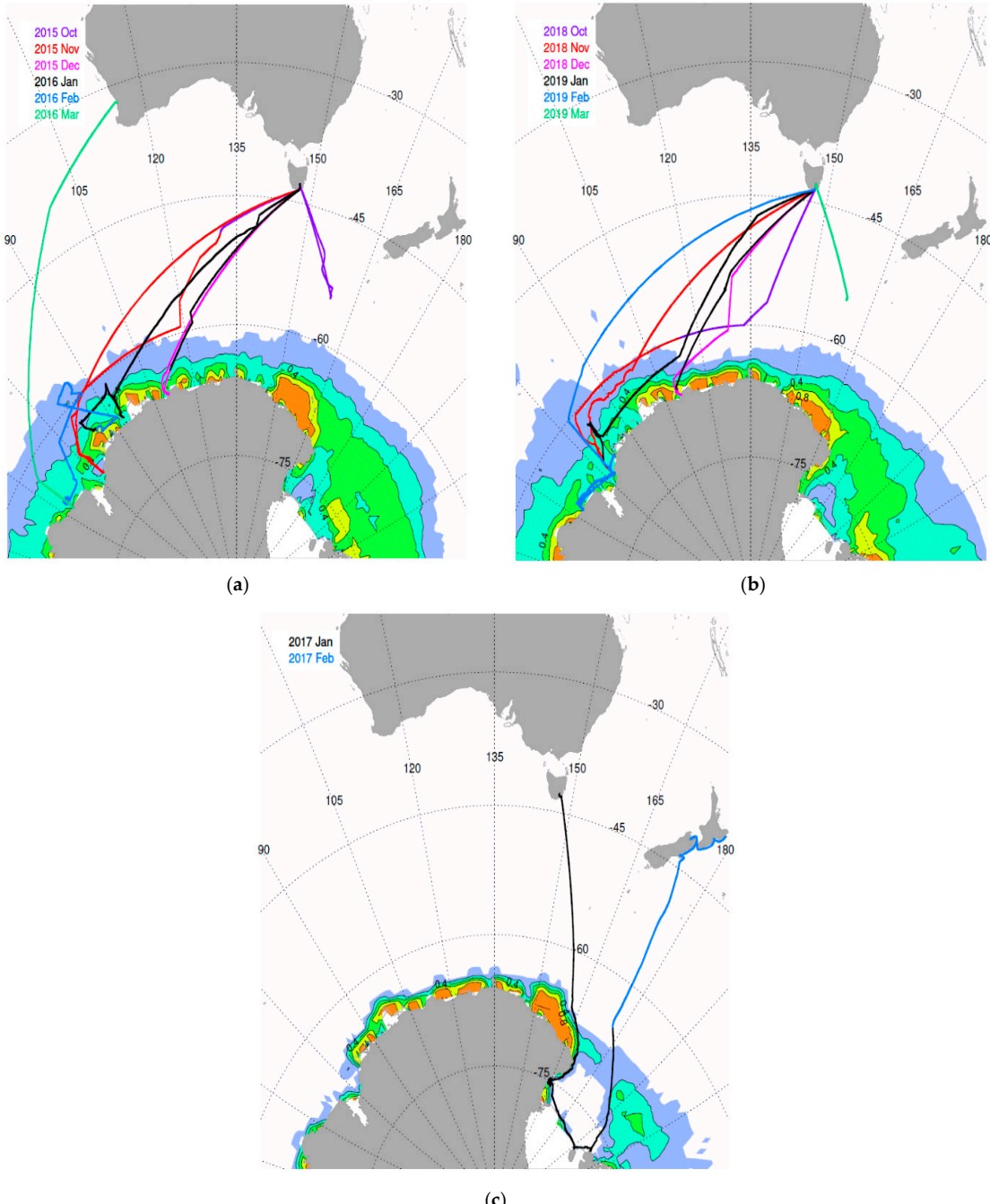

**Figure 1.** Colour-coded ship tracks for each month during the voyages of (**a**) RSV Aurora Australis over the period October 2015 to March 2016, (**b**) RSV Aurora Australis over the period October 2018 to March 2019 and (**c**) MS The World over the period of January to February 2017. The contours of sea ice fraction (0.2, 0.4, 0.6, 0.8, 1.0) are overlaid in different colours based on NOAA Optimum Interpolation Sea Surface Temperature version V2 data. The sea ice concentration data have been averaged over November 2015 to February 2016 for (**a**), over November 2018 to February 2019 for (**b**), and over January 2017 for (**c**).

The primary cloud imager on MS The World was a "PanoView" 4K 360° panoramic camera which is based on a 1/3-inch OmniVision OV4689 four-megapixel complementary metal-oxide-semiconductor (CMOS) sensor and used a 1.1 mm F2.0 lens to produce a 220° field of view (Figure 2b). Images were recorded in the lowest recording resolution mode of 720 × 576 pixels. An advantage provided by the cloud imagers is that a direct view of the sky conditions at the site of the radiation measurements were

available each minute, so effects such as rain, snow, sea spray, and shadowing could be evaluated to aid in quality control of the data.

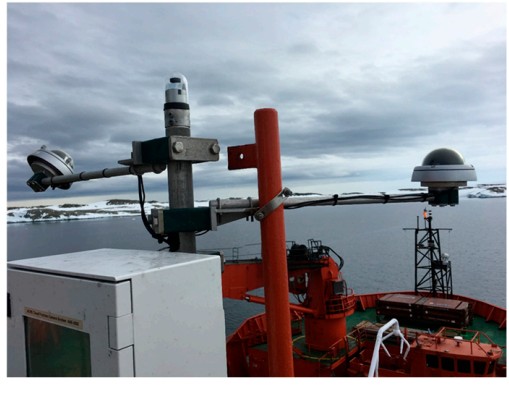 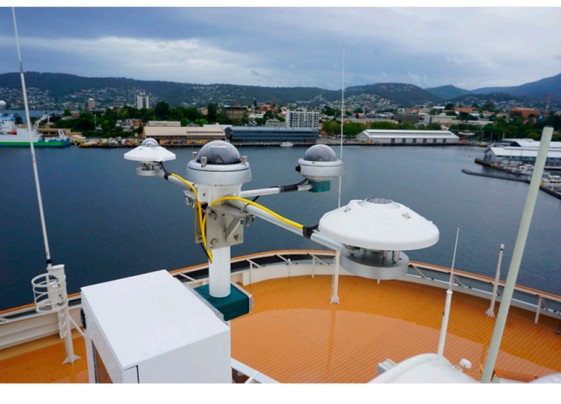

(**a**)                                        (**b**)

**Figure 2.** (**a**) Instrumentation on the "monkey island" of the RSV Aurora Australis during the 2018 to 19 season. The all-sky camera was mounted on the top of the mast. A forward-facing wide-angle camera on the port (left) side was used to provide additional information on surface conditions. An upward-looking wide-angle sky-viewing camera on the starboard (right) side was not used in the analysis presented here. (**b**) Instrumentation on the upper superstructure of MS The World. The wide-angle camera is mounted in the centre of the mast. A second sky-viewing camera with a narrower field-of-view was mounted towards the bow (front) of the ship but was not used for the analysis presented here. The pyranometer is mounted on the arm to the left (toward the port side of the ship), and the pyrgeometer is on the arm to the right (toward the starboard side of the ship).

As the aim of the research was to investigate the influence of local conditions on the radiation environment during daytime, we used a version of common blue-red pixel ratio and differencing methods to differentiate clear sky and cloudy pixels, and sum pixel counts to determine a cloud fraction for each image recorded during the day (solar elevation > 5°). Variants of these methods have been described previously [34–38]. For our pixel ratio method (blue channel divided by red channel; BdR) a threshold of 1.3 was applied to the 8-bit (0–255) blue/red components to distinguish blue (clear sky) pixels. The threshold is camera-dependent (determined by image processing and colour saturation settings) and tailored for general sky conditions (depending on sky brightness, Rayleigh scattering and van Rhijn effects) and was set based on visual inspection of a wide range of images to provide a reasonable discrimination over clear, cloudy and overcast conditions [31]. Figure 3 shows an example of the image processing for two different cameras. Raw images are shown on the left. Detected clear sky pixels are false coloured using a red-yellow scale overlaid on the image on the right. The saturated pixels (all red-green-blue (RGB) channels > 245) and black pixels (all RGB channels < 5) were not included in the pixels counts and are false coloured lime green and brown, respectively. For comparison, the blue-red pixel difference metric (BmR) was also adopted, and pixels with BmR over a threshold of 30 were classified as clear-sky pixels. Those pixels classified as clear-sky by the BmR method but not detected by the BdR method are false coloured using a red-purple colour scale. As in Klekociuk et al. [31], we evaluated the cloud fraction (as the fraction of pixels containing cloud in the total available pixels unaffected by saturation or containing parts of the ship) in a "zenith" region (the 8° diameter region centred in the zenith) and an "all-sky" region.

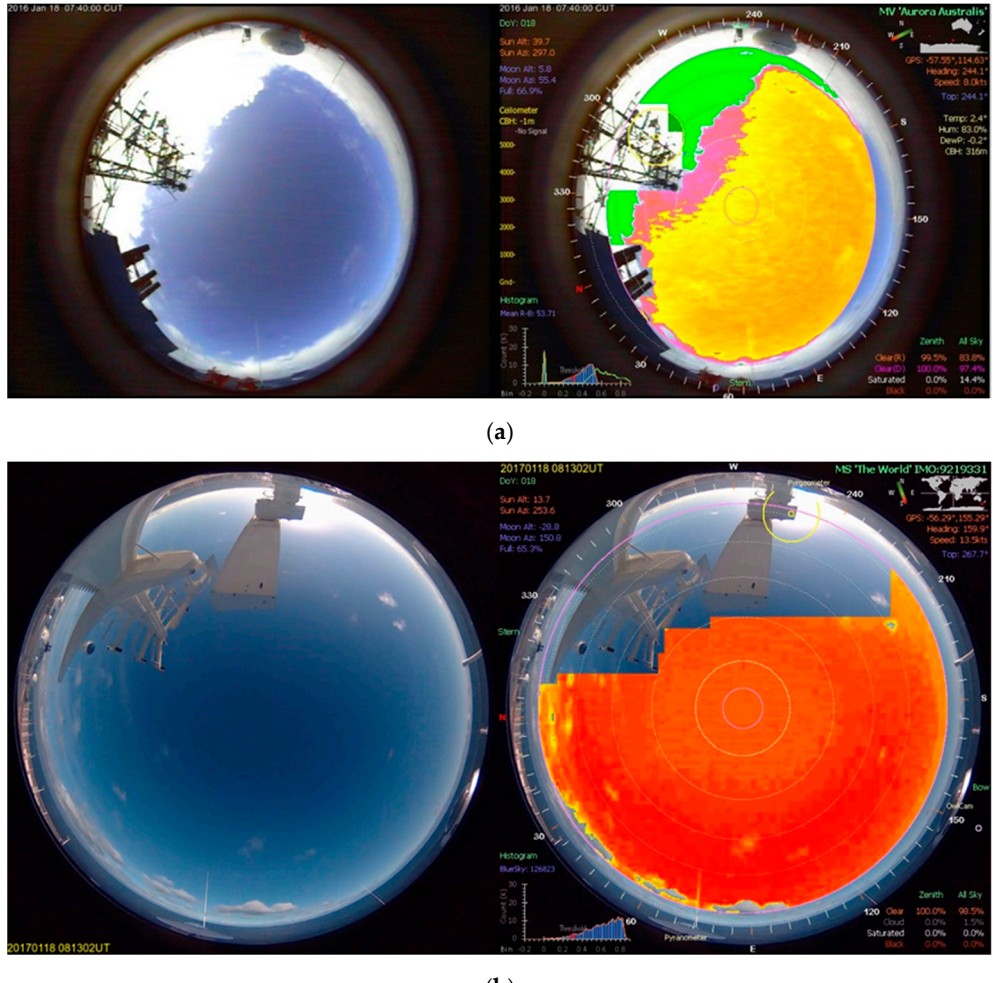

**Figure 3.** Raw and processed images for (**a**) 2015 to 2016 RSV Aurora Australis voyages and (**b**) 2017 MS The World voyage when the sun is behind the ship's superstructure.

*2.3. Radiometers*

The radiation measurements were made with two main arrangements of sensors. For the RSV Aurora Australis voyages during the 2015 to 2016 season, and for the MS The World Ross Sea voyage, we used a Kipp and Zonen model CMP21 pyranometer, sensitive over 285 to 2800 nm [39] and a Kipp and Zonen model CGR4 pyrgeometer, sensitive over 4.5 to 42 μm [40]. Measurements were recorded as 1-min means and standard deviations from 0.1-s-interval observations by a Campbell Scientific CR3000 data logger [41]. Both sensors have a 180° field-of-view.

On RSV Aurora Australis, we also made use of radiation data from the ship's Underway logging system [42]. These data were provided by pairs of Middleton EQ08 pyranometers (sensitive over 300 to 3000 nm; [43]) and Eppley model PIR pyrgeometers (sensitive over 4 to 50 μm; [44]), which were positioned on the starboard and the port bridge wings of the ship. The data consisted of 1-min mean values. During the 2018 to 2019 season, we did not have available the Kipp and Zonen sensors used in the 2015–2016 season but instead used the Underway radiation measurements. A comparison of the different sensors used on RSV Aurora Australis is provided in Table B1 of Klekociuk et al. [31].

The observations on MS The World were made with a different set of Kipp and Zonen sensors to those used on RSV Aurora Australis during the 2015 to 2016 season. Meteorological data during the voyage were obtained using a commercial weather station. A key advantage of these measurements was that that they extended to much higher latitudes and further east than possible from the RSV

Aurora Australis voyages. In particular, the voyage set an official record for the most southerly location for any ship.

### 2.4. Meteorological Reanalysis

For comparison with the ship-based measurements, we obtained the surface data from the fifth major global reanalysis produced by the European Centre for Medium-Range Weather Forecasts (ECMWF) Earth System model (IFS), cycle 41r2 (ERA5) [45]. ERA5 is the newest version of ERA-Interim, which provided a reliable representation of the prevailing meteorological conditions in high latitudes of the Southern Hemisphere [46]. The cloud and radiation fields in the reanalysis are modelled based on physical conditions including trace gas amounts, temperatures, and top-of-atmosphere spectral content. We obtained hourly ERA5 gridded data at 0.25° horizontal resolution for downwelling shortwave and longwave radiation at the surface and their theoretical clear-sky equivalents, as well as total cloud fraction and cloud base height (CBH) [47]. The irradiance values are provided as accumulations over the preceding hour, while the other quantities are provided as instantaneous values. Linear interpolation was applied to each reanalysis field to obtain comparison values for the ship-based measurements at 1-min intervals, using the mid-time of the accumulation as the temporal ordinate for the radiation quantities.

### 2.5. Sea Ice Information

A key issue in surface measurements of cloud and radiation is to obtain statistics that provide an adequate representation of conditions covering different climatological situations. For ship-based observations, the most detailed sea ice information is available from images acquired by a variety of cameras, from which the main quantitative information is the presence or absence of ice cover. The albedo effect of sea ice varies considerably for different amounts of snow cover, making it difficult to decide on a specific value for sea ice albedo. As no specialised instruments for determining sea ice concentration were deployed on these ships during the voyages, we linearly interpolated daily sea ice concentration data from the National Oceanic and Atmospheric Administration (NOAA) Optimum Interpolation (OI) Sea Surface Temperature (SST) version V2 [48] dataset (1° horizontal resolution) to the location of each vessel at 1-min intervals. We defined the observations with sea ice concentration < 0.2 as open-ocean conditions while observations ≥ 0.2 as sea-ice conditions. The threshold we used is the same as for Fitzpatrick and Warren [13] to allow for a direct comparison. We verified the presence or absence of sea ice using images from the cameras described in Section 2.2.

For the two seasons of RSV Aurora Australis, most of the time the vessel was sailing in the open ocean, with sea ice encountered generally south of 59° S. For the voyage of MS The World, sea ice was not encountered until 71° S. Figure S1 shows the distribution of sea ice for each season of voyages. The frequency of sea ice occurrence for the 2015 to 2016 and 2018 to 2019 voyages much higher than for the single 2017 voyage which was made relatively late in the summer.

### 2.6. Measurement Intervals and Averaging

For the RSV Aurora Australis voyages, we analysed ship-based measurements between 22 October 2015 (day of year (DOY) 295) and 21 February 2016 (DOY 52, after which data collection ended due to difficulties with the ship's systems), and 25 October 2018 (DOY 298) and 6 March 2019 (DOY 65). Measurements for the voyage of MS The World were used between 12 January 2017 (DOY 12) and 3 February 2017 (DOY 34).

Following Klekociuk et al. [31], we use the 1-min average radiation data and other underway data to investigate the radiation environment over the three voyages. According to Fitzpatrick and Warren [13], the ship tilting and the deviation of the radiation sensor from a cosine response is greater when the solar zenith angle is over 80° lead to higher uncertainties in the observation of radiation. According to Cronin [49], for cloudy conditions, the albedo approximately decreases with the increase

of cosine of solar zenith angle until the cosine of solar zenith angle is over approximately 0.2. For our analysis, observations with solar zenith angle over 80° were removed.

*2.7. Models for Clear Sky Radiation*

In calculating CRE using Equation (1), we first modelled the clear-sky radiation environment to obtain $S_0(\theta)$ and $L_0(\theta)$. Due to the different voyages and environmental factors (such as the individual sensitivity of the sensors, and their placement relative to surrounding superstructure), we analysed the clear-sky radiation environment for each season separately. To conservatively select the clear-sky cases, we filtered the dataset based on cloud fraction (zenith cloud fraction < 0.3) and irradiance variability (requiring the 1-min standard deviation to be less than 1% of the mean for the Kipp and Zonen sensors, and having a five-sample standard deviation less than 5% of the mean for the Underway sensors). We then visually inspected the corresponding camera images and rejected cases where clouds were obviously visible, or the sun was obscured by the ship's superstructure.

The retained clear-sky short-wave measurements were evaluated in comparison with the total downwelling solar irradiance provided by the R subroutine "insol" [50,51], version 1.2.2. As input to the model we used the observed temperature and humidity, the surface albedo $\alpha$ as used in Equation (4) and an assumed ozone column thickness (at standard temperature and pressure) of 3 mm. We separately regressed the observed SW irradiance against the modelled SW irradiance for each measurement season. There was excellent consistency between the measurements and model over a wide range of solar zenith angles in both open ocean and sea-ice conditions. Details of the regressions are provided in Table 1. Differences in the slope of the regression are ascribed to the influence of the ship's superstructure on the amount of diffuse irradiance measured. From camera images, the fraction of sky obscured for the irradiance measurements was previously estimated as 12% and 3% for the Kipp and Zonen and Underway sensors, respectively on RSV Aurora Australis [13,31], and we obtained 15% for MS The World. The coefficients in Table 1 for RSV Aurora Australis are consistent with the expected obscuration by the superstructure of the diffuse solar clear-sky radiation (which was typically 10 to 30% of the total irradiance depending on solar zenith angle and latitude). This demonstrated that the modelled global clear-sky irradiance did not have an obvious bias (under the reasonable assumption that the sensor calibrations were accurate). The regression coefficient in Table 1 for MS. The World is greater than unity; this is because the superstructure produced an effective enhancement in the diffuse irradiance because of its relatively high albedo (as can be seen in comparison with RSV Aurora Australis in Figure 3). We checked clear-sky situations when the sun was directly behind the ship's structures as seen from the pyranometer. The measured irradiance in this situation was of the diffuse component. This was on-average enhanced by a factor of 1.02 relative to the modelled diffuse radiation. As this was similar to the enhancement in the case of direct sun, we expect all measurements to be enhanced, with the coefficient provided in Table 1 providing an upper limit to the magnitude of the effect. In order to correct for the separate effects of the surrounding superstructure in each season, we divided the measured short-wave irradiance by the relevant coefficient m in Table 1 to provide a scaled value. This was done under the assumption that the modelled global clear-sky irradiance provided an appropriate estimate of $S_0$. The residuals from the regressions are shown in Figure 4a.

**Table 1.** Results of linear regressions ($y = m \times x$) to clear-sky global modelled ($x$) and observed ($y$) SW irradiance. The Pearson linear correlation coefficient, R, is also provided.

| Season | Vessel [1] | m | Number of Measurements | R |
|---|---|---|---|---|
| 2015–2016 | AA | 0.9574 ± 0.0008 | 28 | 0.999 |
| 2017 | TW | 1.0356 ± 0.0019 | 153 | 0.999 |
| 2018–2019 | AA | 0.9863 ± 0.0008 | 41 | 0.999 |

[1] AA = RSV Aurora Australis, TW = MS The World.

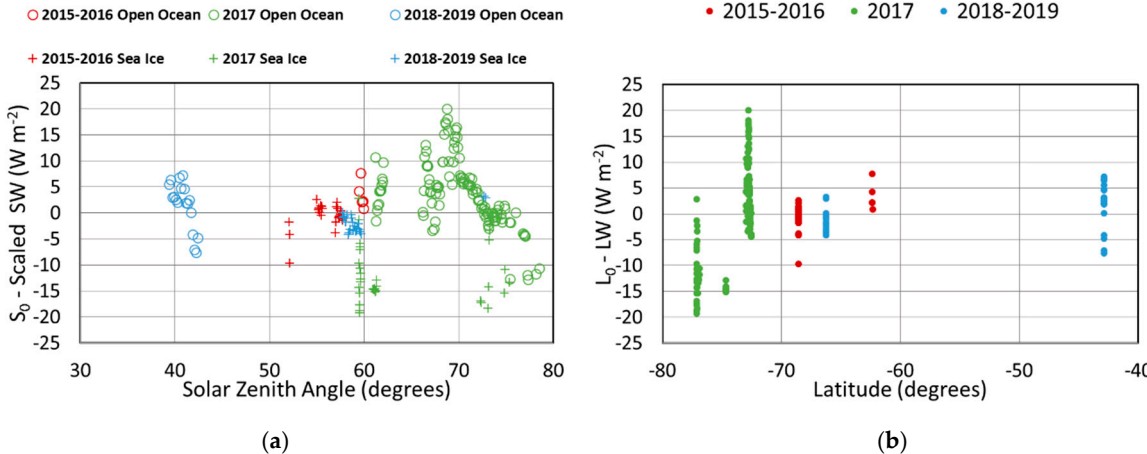

**Figure 4.** Residuals for clear-sky radiation models for each season. (**a**) $S_0$—scaled observed SW irradiance (the observed SW irradiance divided by the regression coefficient m in Table 1) as a function of solar zenith angle. Distinction is made between conditions of open ocean (open circles) and sea ice (crosses). (**b**) $L_0$—observed LW irradiance as a function of latitude.

For the clear-sky cases, we used the model of Ilango [52] to provide an estimate of the LW irradiance using the observed humidity and air temperature. We found that this model provided a reasonable approximation to the observations, which was improved by a polynomial regression against the observations. The residuals from the regression are shown in Figure 4b as a function of latitude. The equation used to model the clear-sky LW irradiance was:

$$L_0 = -0.00303(39) \times L^2 + 2.63(21) \times L - 231 \tag{4}$$

where L is the irradiance obtained from the clear-sky LW model [52], and the figures in parentheses are standard errors in the last two digits of the coefficients (222 measurements, $R^2 = 0.96$).

We selected the set of measurements for which the radiation sensor data would not have had the sun placed behind surrounding superstructure as determined from analysis of the initial set of clear-sky candidate measurements. This approach ensured that cases of thin or broken cloud were not affected by shadowing. These measurements were then used for the calculation of CRE, SCE, LCE (Equations (1)–(3)) using $\varepsilon = 0.97$ and $\alpha = 0.055$ for open ocean to maintain a consistent comparison with previous studies [30,31]. The ratio of observed SW to clear sky conditions at the same solar zenith angle, termed the raw cloud transmittance (TRC), is also calculated using the voyage data [13]:

$$TRC = S(\theta)/S_0(\theta) \tag{5}$$

with TRC = 1 for clear-sky conditions.

## 3. Results

### 3.1. Meridional Variation of CRE

The distribution of CRE for each latitude band is shown in Figure 5. From 40° S to 55° S, the mean CRE becomes more negative. This reverses as latitudes continue increasing from 55° S to 80° S. The transition at 55° S coincides with the crossing of the oceanic polar front (the Antarctic convergence) where colder (warmer) waters are found poleward (equatorward) of the front. The density curves show monomodal distributions from 40° S to 70° S but bimodal distributions appear for latitudes greater than 70° S where a higher percentage of observations with less negative CRE (clearer skies) occur. The mean CRE for the three summer observations was −157 ± 100 W m⁻², which indicates that clouds over the Southern Ocean had a cooling effect on the sea surface. Here, and for other

observed mean irradiance values provided below, the error quoted is ± 1 standard error from formal propagation of uncertainties in $S_0$ and $L_0$ which dominate the uncertainty budget. The mean compares with a climatological average over 30° to 60° S of approximately −84 W m$^{-2}$ derived from Figure 14 of Haynes et al. [2].

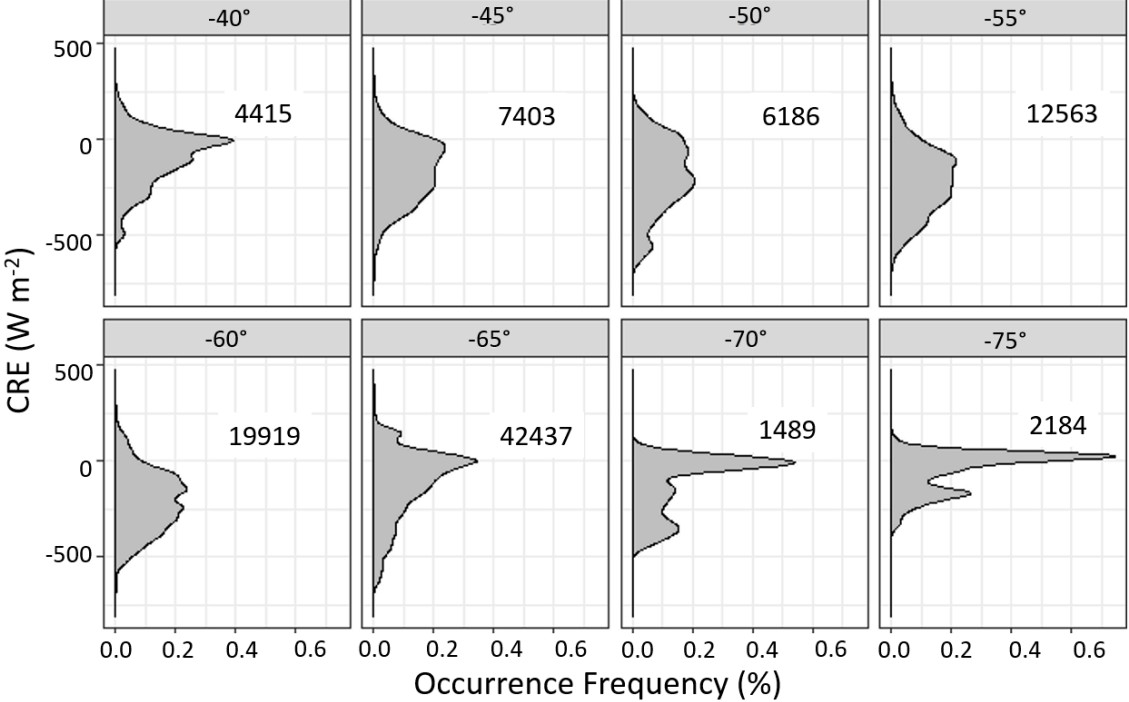

**Figure 5.** Histograms of 1-min averaged cloud radiative effect (CRE) as a function of percentage occurrence for different latitudes in 5-degree bands. The data are from all voyages. The latitude of each band-centre (degrees) is indicated at the top of each plot and the total number of 1-min observations in each latitude band is indicated in each panel.

### 3.2. Cloud Fraction and CRE

Sky conditions were generally cloudy or overcast over most of the observation time. The daily average cloud fraction for the 2017 observations (0.65, standard deviation σ = 0.42) from MS The World was lower than in the 2015 to 2016 (0.75, σ = 0.26) and 2018 to 2019 (0.73, σ = 0.29) seasons, which were made over a wider area of the Southern Ocean. On average, the observed all-sky cloud fraction for the 1-min observations over three different seasons was 0.74 (σ = 0.29) which was less than the value of 0.96 found by Klekociuk et al. [31] for a subset of the 2015 to 2016 measurements over the latitude range 57.5° S to 66.6° S. While the measurements analysed by Klekociuk et al. [31] were made near latitudes where the cloud fraction is highest, they were concentrated in a relatively confined oceanic region and short time-span that was potentially not fully representative of the broader characteristics of the region covered in the present study.

Figure 6 shows the average all-sky cloud fraction in 5-degree latitude bands for each voyage and for the sum of three sets of data. In general, the latitude distribution is monomodal, with average cloud fraction peaking around 60° S, with steeper reduction towards polar latitudes than toward mid-latitudes. In Figure 6e, the highest cloud fraction measurements (cloud fraction over 90%) show a strong latitudinal dependence. These occurrences are most common around 60° S and are significantly rarer poleward as clear sky conditions begin to dominate. The 2015 to 2016 and 2018 to 2019 voyages of RSV Aurora Australis had similar routes, however, the 2017 voyage route of MS The World was further to the east and to higher latitudes, and shows a different distribution (in Figure 6, compare panels (a) and (c) with panel (b)). The dips at 70° S in panels (b) and (d) of Figure 6 are due to the

generally cloud-free conditions in the Ross Sea during the period from DOY 22 to 25 in 2017. However, a generally higher percentage of cloud fractions of 0.3 or less are apparent in Figure 6e south of 60° S, which also includes measurements from the RSV Aurora Australis. Note also that in summer, the cloud fraction in the Ross Sea region (near 180° E at the edge of Antarctica) is similar to other coastal regions of East Antarctica (see Figure 5 of Listowski et al. [24]), and this results in a general levelling off of values towards higher latitudes in Figure 6d.

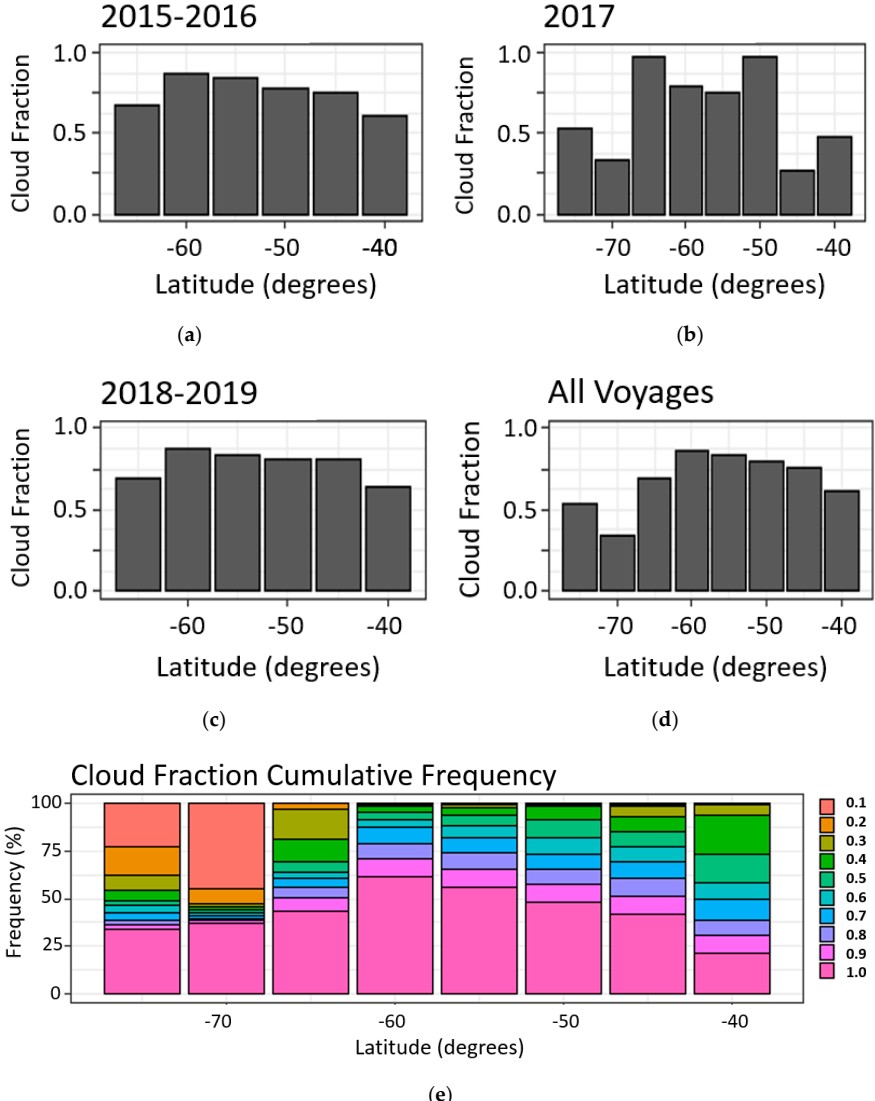

**Figure 6.** The average cloud fraction (%) in 5-degree latitude bands for (**a**) the 2015 to 2016 voyages, (**b**) the 2017 voyage, (**c**) the 2018 to 2019 voyages, (**d**) over all voyages and (**e**) over all voyages but shown as a cumulative frequency distribution.

In Figure 7, the frequency distribution of cloud fraction is shown for the observations and the interpolated ERA5 data. Over 50% of the observations had a cloud fraction greater than 0.9 and only 5% of the observations had a cloud fraction less than 0.2, which is similar to the results of Fitzpatrick and Warren [13]. In comparison, ERA5 showed a positive bias for the occurrence of cloud fractions less than 0.3, and a negative bias for cloud fractions of 0.3 and larger, including overcast conditions. Overall, the 1-min mean observed cloud fraction of 0.75 ($\sigma = 0.28$) was slightly higher than 0.70 ($\sigma = 0.35$) obtained from the interpolated ERA5 data.

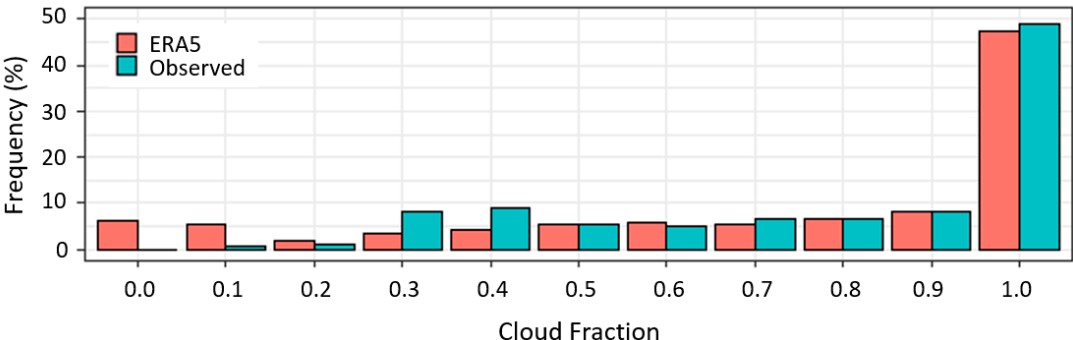

**Figure 7.** Histogram of cloud fraction for all observations (red) and the corresponding interpolated data from ERA5 (cyan) in 0.1 bins.

Other reanalyses have shown a similar tendency to underestimate cloud fraction in the Southern Ocean and Antarctic region. Kuma et al. [28] found that the Modern-Era Retrospective analysis for Research and Applications (MERRA), Version 2 (MERRA-2) underestimates Southern Ocean cloud cover by 18% (see their Figure 7 and Section 5.2). Naud et al. [53] compared cloud cover from the Interim ECMWF reanalysis (ERA-Interim, which preceded ERA5) and Version 1 of MERRA against satellite measurements. ERA-Interim agreed within 5% but MERRA had a significant underestimate. Although from a latitude-restricted temperate region of the Southern Ocean, Protat et al. [30] in their Figure 1b showed a similar radar-observed cloud fraction to our assessment, except an increase at cloud fractions below 0.1 (which may be a characteristic of the temperate region examined or the short duration of their dataset).

Figure 8 shows the relationship between cloud fraction LCE, SCE, and CRE. In general, LCE is relatively insensitive to the presence of cloud at cloud fractions of 0.4 or lower (as suggested by the relatively flat response in this part of Figure 8a). LCE increases for cloud fractions between 0.4 and 0.7 and then tapers towards the highest cloud fraction. The response in SCE (Figure 8b) is generally complementary to the LCE response, although the absolute magnitude of the dependence on cloud fraction in SCE is larger by a factor of about five. For cloud fractions between 0.1 and 0.3, broken cloud tends to enhance the diffuse component sufficiently to negate any attenuation of the direct component and the SCE for this range is near zero. Some indication of this enhancement is also apparent in Figure 6 of Key and Minnett [8] which shows the variation of SCE with cloud amount during an Antarctic spring voyage. For cloud fractions greater than 0.3, SCE becomes progressively more negative. Overall, CRE shows positive values between cloud fractions of 0.1 and 0.3 and negative values for other cloud fractions.

*3.3. Effect of Sea Ice on CRE*

Sea ice reduces outgoing LW radiation by decreasing the transport of heat and moisture between the ocean and atmosphere, and by virtue of its higher albedo increases the upward scattering of incoming SW radiation. In this section, we compare and contrast CRE estimates made in open ocean conditions and in the presence of sea ice.

Figure 9 shows the histograms of TRC for open ocean and sea ice conditions. Both histograms show a bimodal distribution, with the peak for TRC < 1 over open ocean shifted to lower values than in the sea ice region, and a tendency for a higher occurrence of TRC > 1 over sea ice. Visible in Figure 9 is the Köhler gap, which is described by Fitzpatrick and Warren [13] as the sparsely populated region of observed TRC between the peaks of the bimodal distribution, and its presence is due to the clean atmosphere with low aerosol loading over the Southern Ocean. The Köhler gap can also be seen in Figure S2, which shows density distributions of TRC as a function of solar zenith angle over open ocean and sea ice, at TRC ~ 0.8. The observations with TRC values over 1.0 in Figure 9 are due to the enhancement of diffuse SW due to the presence of broken clouds [13].

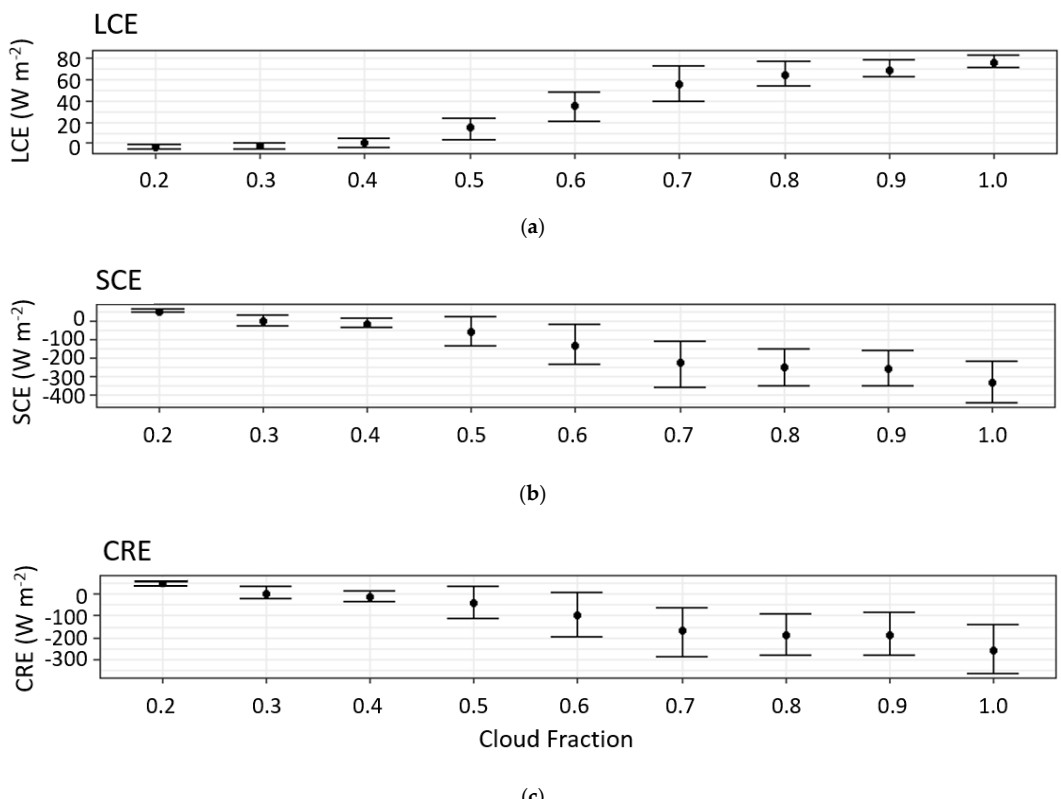

**Figure 8.** (**a**) Long-wave cloud radiative effect (LCE), (**b**) short-wave cloud radiative effect (SCE), and (**c**) net cloud radiative effect (CRE) versus cloud fraction obtained from all observations. Medians are shown as filled dots and the bars span the interquartile range. The data are averaged in intervals of 0.1 and are centred at the upper limit of each bin (for example, the bin at 0.1 is the average for 0.0 to 0.1).

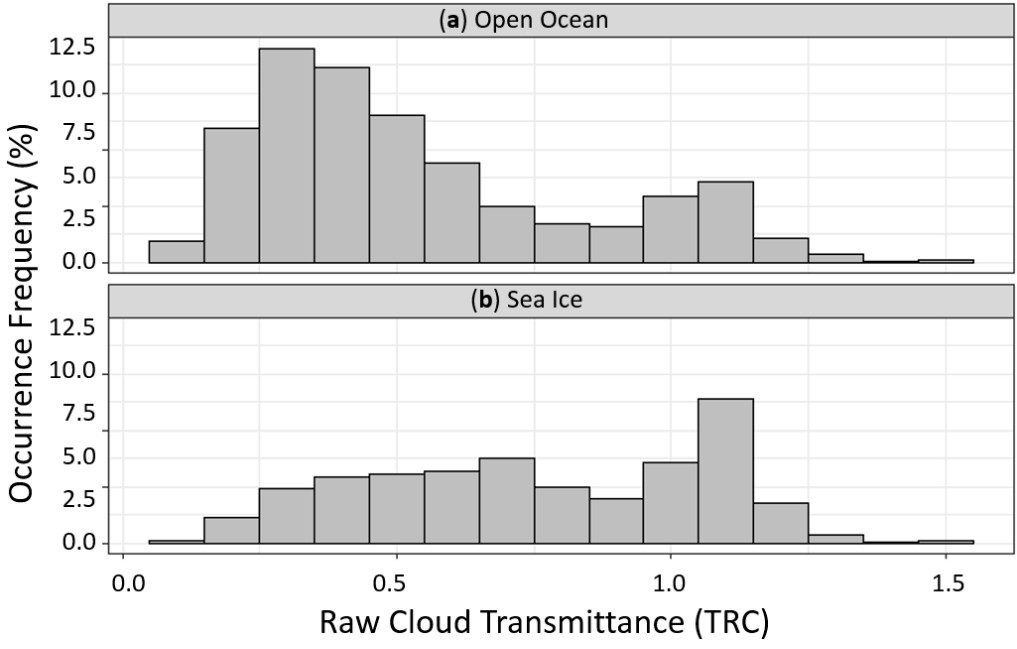

**Figure 9.** Frequency distribution of raw cloud transmittance (TRC) in percentage over (**a**) open ocean and (**b**) sea ice. Data are from all voyages.

Over open ocean, TRC is predominantly under 0.5, but over sea ice, there are observations around 1, which indicates that clear sky was more common over sea ice than over the open ocean. We also calculated the averaged TRC for different combinations of sea conditions and sky conditions to investigate the sensitivity of TRC to the presence of sea ice in under clear sky and cloudy sky. For clear or generally cloud-free conditions (cloud fraction < 0.2), the average TRC over sea ice was 5% higher than over the open ocean. For cloudy conditions (cloud fraction ≥ 0.2) the average TRC over sea ice was 37% higher than over the open ocean. The influence of sea ice on increasing the reflection of SW is therefore amplified with cloud presence, mainly due to the multiple reflections of the high-albedo sea ice surface and clouds.

The frequency distribution of CRE as a function of the solar zenith angle is shown in Figure 10 for open ocean and sea ice conditions. For the open ocean, there is a general tendency for the more negative values of CRE to have a stronger dependence on solar zenith angle than for sea ice conditions. The lowermost boundary of the points in the panels of Figure 10a are similar to that obtained by Minnett [54] for a range of cloud types in the Arctic. Figure 10b separately shows the average CRE for open ocean and sea ice conditions in 5° solar zenith angle intervals. This shows that the CRE in the presence of sea ice is higher than for open ocean conditions. There is an approximate linear relation between CRE and solar zenith angle over open ocean under cloudy conditions which is less apparent over sea ice. For solar zenith angles of 50° to 60° a relatively concentrated amount of observations were made in the sea ice zone under generally cloud-free conditions, and this tends to make the average CRE significantly less negative than for the open ocean in this range of solar zenith angles. Additionally, the presence of icy rather than liquid cloud particles at the higher latitudes will tend to reduce the SCE by making the clouds more transparent [7]. When weighted by the duration at which the solar zenith angle is between the range for each bin shown in Figure 10b we obtained an average net CRE for observations over the open ocean as $-199 \pm 105$ W m$^{-2}$ and over sea ice as $-113 \pm 92$ W m$^{-2}$.

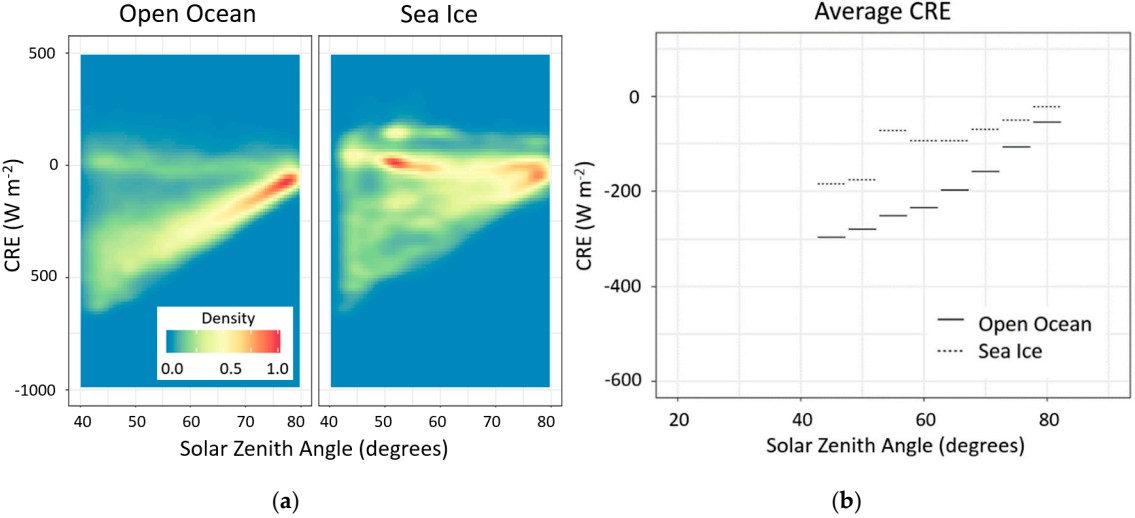

**Figure 10.** Distribution of net cloud radiative effect (CRE) versus solar zenith angle for (**a**) open ocean (left) and sea ice (right) with frequency of occurrence (density) scaled to 0–1 (blue to yellow to red). (**b**) Average CRE versus solar zenith angle in 5-degree intervals over open ocean (solid lines) and over sea ice (dashed lines). Data are from all voyages.

### 3.4. Comparison with ERA5 Radiation Data

As noted in Klekociuk et al. [31], the spectral sensitivity of the irradiance sensors allows direct comparison with the ECMWF radiation model. The ERA5 SW irradiance is plotted as a function of the observed SW irradiance in Figure 11a, which shows that majority (70%) of the points are distributed above the one-to-one line, indicating that ERA5 predominantly overestimates SW irradiance. The mean

observed 1-min SW irradiance was 80 W m$^{-2}$ less than for ERA5, while the observed daily average was 60 W m$^{-2}$ less than for ERA5. This is similar to SW bias in ERA-Interim reported by Klekociuk et al. [31], and for the Antarctic region by Zhang et al. [55]. When the observed SW irradiance was larger than approximately 700 W m$^{-2}$, there was a tendency for points to be located below the one-to-one line (Figure 11a). This could occur if ERA5 tends to underestimate radiation enhancement under broken cloud, or if ERA5 overestimates the transmittance of thin cloud. At low SW irradiance (below about 500 W m$^{-2}$), ERA5 was consistently higher than the observed values. According to Cronin [49], the scattering of downward SW irradiance is significantly amplified with increasing solar zenith angle in the presence of low-level clouds. Furthermore, when the sun is low, more incoming SW is absorbed when the incoming SW passes through a longer path length in the clouds. Also, loss of SW is related to the apparent increase in cloud fraction towards the horizon due to perspective. Hence, the positive bias by ERA5 at low values of SW irradiance could be due to an underestimation of cloud transmittance, cloud thickness or the consequences of perspective for broken cloud at low solar zenith angles.

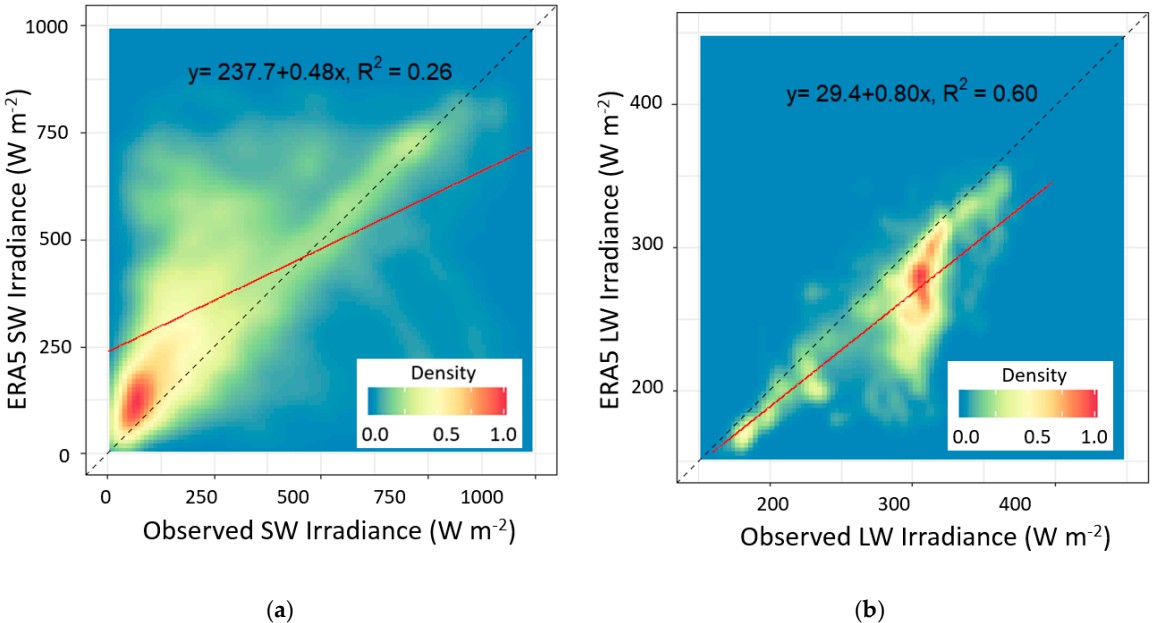

(**a**)　　　　　　　　　　　　　　　　　　　　　　　　　(**b**)

**Figure 11.** Comparisons of observations from ship-based sensors with ERA5 data interpolated to the location of the ship with occurrence frequency (density) scaled to 0–1 (blue to yellow to red). (**a**) SW irradiance. (**b**) LW irradiance. The scatterplots show linear regression equations and R$^2$ correlation coefficients between the associated horizontal (x) and vertical (y) ordinates. The regression lines (solid red) and one-to-one line (dashed black) are also shown. Both the observed data and ERA5 data are at 1-min intervals.

ERA5 underestimated the LW irradiance for 90% of the observations (points distributed below the dashed 1-to-1 line in Figure 11b). The mean observed LW irradiance was 288 ± 105 (σ = 43) W m$^{-2}$ which is 30 W m$^{-2}$ higher than the mean LW irradiance from ERA5 of 258 (σ = 44) W m$^{-2}$. The daily averaged observed LW for was 295 ± 108 (σ = 38) W m$^{-2}$, which is 31 W m$^{-2}$ higher than the mean for ERA5 of 264 (σ = 41) W m$^{-2}$. The relative difference is relatively small (observed ~12% greater than ERA5) and within the uncertainty limits, and similar to the bias for all conditions obtained by Silber et al. [56] at McMurdo and WAIS Divide in Antarctica.

For observed clear-sky cases (where we also required the ERA5 cloud fraction to be zero), the mean of observed – ERA5 for the LW irradiance was +1 (σ = 10) W m$^{-2}$ (153 observations; median −6 W m$^{-2}$, interquartile range 16 W m$^{-2}$) which is consistent with the near-zero mean clear sky LW bias obtained by Silber et al. [56]. In the case of the clear-sky SW irradiance, the mean of observed – ERA5 was +25 (σ = 36) W m$^{-2}$ (median +26 W m$^{-2}$, interquartile range 19 W m$^{-2}$). We did not have a suitable

range of observing conditions to determine if this difference was sensitive to solar zenith angle, surface conditions or latitude, but this difference potentially relates to the transparency of the atmospheric column modelled in ERA5 being lower by about 7%. We also examined differences when the cloud fraction was less than 0.3 in both the observations and ERA5. Histograms of the differences are provided in Figure S3. The medians of the SW and LW differences are $-12$ W m$^{-2}$ and $+14$ W m$^{-2}$, respectively, with interquartile ranges of 71 W m$^{-2}$ and 12 W m$^{-2}$, respectively. These cases have some influence from cloud, and the differences are consistent with the direction of biases in ERA5 due to underestimation of cloud properties noted above in this section (that is, ERA5 being biased high and low for SW and LW when cloud is observed, respectively). In addition, the results suggest that the SW clear sky irradiance in ERA5 is biased slightly low.

In Figure 12a we show the observed CRE as a function of solar zenith angle. The lower limit of CRE is approximately bounded by a linear function of the cosine of the solar zenith angle, as found by Minnett [54]. There is a tendency for values to lie below the dashed lower boundary in Figure 12a at the highest solar zenith angles, which could be a result of the perspective effect of the greater path length through clouds noted above. The upper range includes positive values, which is generally due to the radiation enhancement from broken cloud or the effects of sea ice when present. Figure 12b shows more clearly the reduced spread of measurements at large solar zenith angles (small cosine of solar zenith angle). In Figure 12a, we also show (as small red dots) 3-h averaged ERA5 data (rather than the interpolated 1-min data used in Figure 12b) which shows a similar distribution to the observations but with two differences. Firstly, values tend to be concentrated towards higher values unlike in the observations which is a consequence of ERA5 generally having less negative CRE. Secondly, the ERA5 data tend not to show positive CRE which could be due to the underestimation of the scattering effects in the presence of sea ice, or alternatively, if the model underestimates the radiation enhancement under broken clouds.

Figure 12a also shows the mean observed values of CRE at specific solar zenith angles for different cloud types. Based on camera images, clouds were manually classified by inspection for solar zenith angles of 69°, 60° and 46° ($\pm 1$°), with the angles chosen for comparison with results of Klekociuk et al. [31]. Four cloud classifications were used: cumuliform cloud (cumulus with blue sky breaks; 33% occurrence), multilayer cloud (clouds in distinct layers; 22% occurrence), precipitation cloud (detected when rain droplets or snow accumulated on the dome of the cloud imager; 18% occurrence) and stratiform cloud (clouds with unbroken sky coverage and generally uniform appearance; 26.5% occurrence). The strongest (most negative) CRE was observed for precipitation clouds and the weakest (least negative) CRE was observed from cumuliform clouds, which is similar to the results in Minnett [54] and also in Klekociuk et al. [31]. The averaged observed CRE for multilayer clouds was similar to stratiform clouds. The multilayer clouds generally occurred where the cloud fraction was > 0.7, and mostly when there were concurrently clouds that appeared to be low based on their appearance and motion; the similarity of CRE with the stratiform type could be due to a dominant effect by the lowermost clouds, or if stratiform clouds are also generally accompanied by higher layers. Minnett [54] found a clearer distinction between the CRE of multilayer and stratiform clouds, with the multilayer clouds having less negative CRE, although cloud classification was based on visual spot measurements that could not be further diagnosed with images. We also examined the difference in the CRE between the observations and ERA5. Generally, the cumuliform clouds showed a negligible difference, encompassing zero in the interquartile range at each of the three zenith angles. The other three cloud types, which in combination are more common over the Southern Ocean, showed an obvious negative difference (positive ERA bias) that increased towards smaller solar zenith angles. We could not distinguish a consistent difference in the bias between these cloud types, but the multilayered and precipitation types generally had the largest bias. A statistical follow-up study is recommended to examine the CRE bias in cases where specific cloud types are simultaneously present in observations and models.

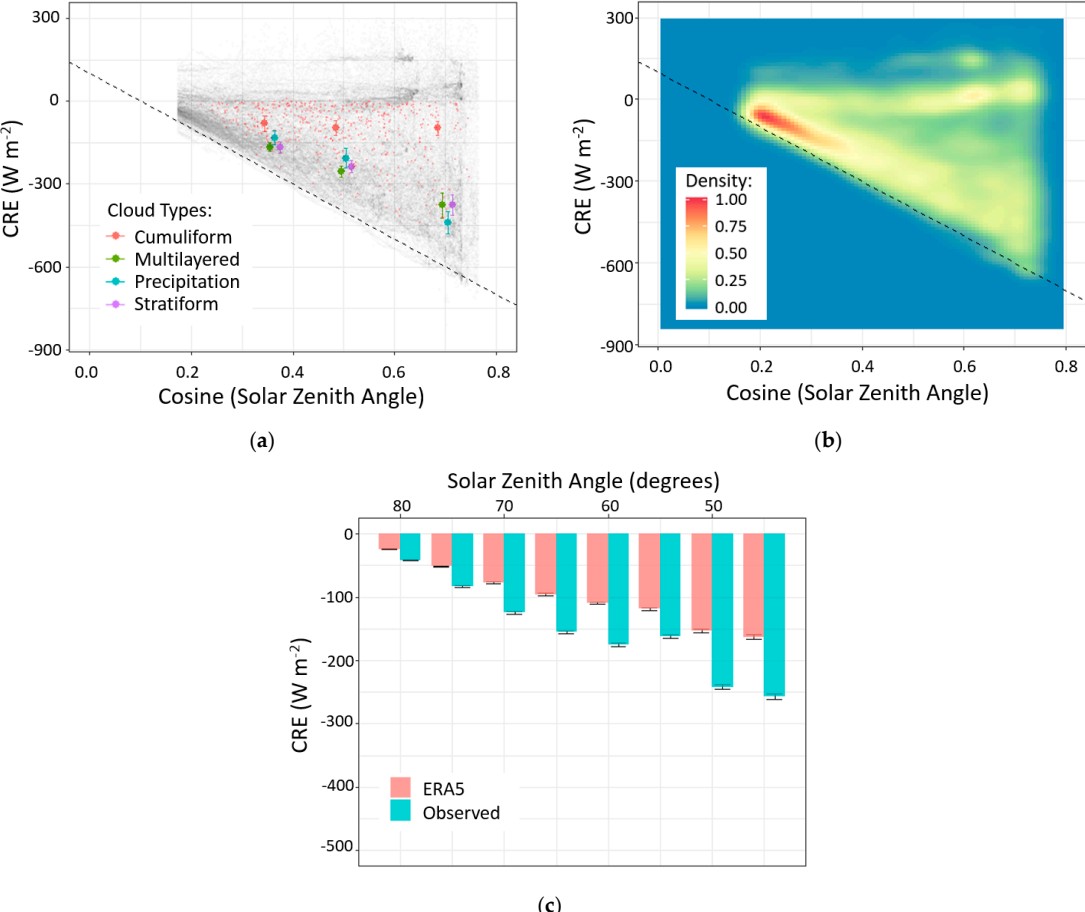

**Figure 12.** Net cloud radiative effect (CRE) from all observations as a function of the cosine of solar
zenith angle as (**a**) individual points (grey filled dots) and (**b**) coloured by occurrence frequency (density)
scaled to 0–1 (blue to yellow to red). In (**a**) small red filled dots show 3-hourly average values from
ERA5; large filled dots are averages coloured according to cloud type inferred from camera images at
three solar zenith angle: 69°, 60° and 46° (values are offset either side of the centre of each angle for
clarity; vertical bars span ±2 standard errors in the mean): cumuliform clouds (orange), multilayered
clouds (green), clouds causing surface precipitation (blue), and stratiform clouds (purple). In (**a**,**b**) the
dashed line shows the approximate lower limit of measured net CRE from Minnett [54]. The observed
1-min CRE values are shown in half-transparent grey dots and 3-hourly averaged ERA5 data (red dots).
(**c**) Histograms of averaged net CRE as a function of solar zenith angle from all 1-min observations (blue)
and averaged net CRE from 1-min interpolated ERA5 data (red). The vertical bars span ±2 standard
errors in the mean.

In Figure 12c, the average net CRE for ship-based observations (blue) is stronger than CRE from
ERA5 (red) for all solar zenith angles. Since the observed data is filtered to avoid being shadowed by
the ship's superstructure, both observed and ERA5 data are weighted by the duration at which the
solar zenith angle is between the range for each bin shown in Figure 12c to calculate the average net
CRE. The average net CRE of ERA5 is −99 (σ = 48) W m$^{-2}$ which is 64 W m$^{-2}$ higher than the observed
average of −155 ± 100 (σ = 73) W m$^{-2}$.

As noted above, a specific radiation bias in ERA5 is not apparent when cumuliform clouds are
observed, at least in the cases selected. We have not examined ERA5 in detail to determine if the model
simultaneously indicates the presence of this type of cloud, but both the observations and model show
a preference for broken cloud at these times. As shown in Figure S4a, the radiation bias in ERA5 is
small when the observed cloud fraction is less than approximately 0.7 in the open ocean and 0.5 in
sea ice. The bias in ERA5 increases with greater cloud fraction which points to the transparency of

the clouds being underestimated in the model. In Figure S4b, the ERA5 cloud fraction is used as the x-ordinate. Generally, the bias in ERA5 is apparent at cloud fractions greater than 0.3; this also indicates that when ERA5 makes sufficient cloud to have a radiative effect, it is too transparent. The lower cloud fraction threshold for the onset of bias in Figure S4b compared with Figure S4a is consistent with ERA5 having a greater occurrence of cloud fractions below this value (apparent from Figure 7).

Further insight is gained from Figure S5. In Figure S5a, we show the occurrence of the CRE difference as a function of the lifting condensation level (LCL) calculated from observed temperature and relative humidity (given in meters by 125 times the difference between the observed air temperature and the dew point temperature). A distinct LCL exists separately from the inversion height when turbulence is insufficient to maintain mixing throughout the entire boundary layer, thus a decoupled layer forms between the LCL and the inversion. This can exist in cloudy and clear-sky conditions (see, for example, Figure 5 in Alexander and Protat [57]). While the CBH can indicate the top of the LCL, the lowest layer cloud may not necessarily be coupled to the surface. Apparent in Figure S5a is that when the LCL is below about 1000 m for both open ocean and sea ice conditions, which represents most of the observations, ERA5 consistently underestimates the CRE by about 100 W m$^{-2}$. This difference is similar to the overall average CRE bias we find for ERA5. As boundary layer thermodynamics controls the LCL, a deficiency of ERA5 in this area could be a specific reason for this CRE difference. Kuma et al. [28] discuss that near-zero LCL is a reasonable indicator of fog over the Southern Ocean (their Figure 8), and we see a high proportion of cases of low LCL in the open ocean (Figure S5b). The CRE difference below 500 m in Figure S5a is similar to or slightly less than at other heights below 1000 m. This suggests that the modelled radiation properties of fog or very low cloud near the surface may be adequate, or even compensating for bias.

We point out that LCL values greater than 1000 m correspond to surface relative humidities less than about 55% which tend to be observed in the lowest and highest latitudes of our study region, though not exclusively. The behaviour in Figure S5a indicates that the ERA5 radiation bias becomes apparent at surface humidities greater than this threshold. We do not find a specific bias in the ERA5 surface relative humidity compared to the ship measurements. This is at least expected in the case of measurements from RSV Aurora Australis which are reported on the Global Telecommunications System and most likely used to inform the ERA5 variational analysis (a similar reasonable agreement was found for a subset of the measurements against ERA-Interim [31]).

When plotted against CBH, the CRE difference shows generally negative values, except adjacent to the surface (Figure S5c), indicating that the CBH is probably not a strong determinant of model bias. In Figure S5c the CRE difference is more strongly negative in the sea ice zone when the CBH is modelled as being between approximately 1200 m and 2000 m. This may relate to specific properties of the modelled population of clouds that appear to have a preference for residing at these heights in the sea ice zone compared with the open ocean (seen in Figure S5d).

## 4. Summary and Conclusions

Using ship-borne measurements of SW and LW radiation and cloud fraction we have examined the radiation environment over the Southern Ocean within the region bounded by 42.8° S to 78.7° S and 62.6° E to 157.7° W over three summer seasons. The measurements were made during voyages of RSV Aurora Australis over the period October 2015 to February 2016 and October 2018 to March 2019, and MS The World over the period of January to February 2017. In the measurement region, the sky was predominantly cloudy, with clear-sky or cloud-free observations (cloud fraction < 0.2) reported less than 5% of the time. The mean SCE is observed to increase while mean LCE is observed to decrease with increasing cloud fraction. The sensitivity of SCE to cloud fraction is approximately five times higher than LCE which suggests that low-level and mid-level clouds dominate the observation region. Total CRE is observed to become more negative with increasing cloud fraction.

Over the Southern Ocean, the mean observed CRE was predominantly negative with a minimum around 55° S, and progressively less negative towards more northern and southern latitudes. More

observations with positive CRE were obtained when close to Antarctica as a result of less cloud cover. The interaction between sea ice and clouds was observed to increase observed SW irradiance and decrease TRC. The Köhler gap was detected in the distribution of TRC which indicates low aerosol loadings over the observation region. Over sea ice, the observed weighted CRE was 107 W m$^{-2}$ higher than over open ocean due primarily to a reduction in the cloud fraction. Overall, compared with the study of Klekociuk et al. [31] over a region of the Southern Ocean near 60° S, our results show a less negative CRE (73 W m$^{-2}$ higher).

Compared with the ERA5 reanalysis, the observed SCE was lower than for ERA5 for 68% of observations, and the daily average observed SCE was 77 W m$^{-2}$ lower than for ERA5. For LCE, the observed value was higher than 75% of ERA5 values, and the daily average observed LCE was 18 W m$^{-2}$ higher than for ERA5. However, conditions with a cloud fraction of less than 0.3 were more prevalent in ERA5. The daily average observed cloud fraction of 0.75 ($\sigma$ = 0.23) was greater than the value of 0.71 ($\sigma$ = 0.27) obtained for ERA5. The distribution of observed CRE as a function of solar zenith angle was more scattered than ERA5 and more concentrated at the lowermost levels. The average CRE weighted in 5-degree solar zenith angle intervals for ERA5 was −99 ($\sigma$ = 56) W m$^{-2}$, which was 64 W m$^{-2}$ higher than the observed value of −155 ± 100 ($\sigma$ = 73) W m$^{-2}$. Therefore, the observed CRE was significantly more negative than for ERA5, which appears related to the greater abundance of clouds over the Southern Ocean than modelled, and potentially to the clouds having lower transmittance than modelled. In terms of clear sky conditions, there was no significant bias in the ERA5 LW irradiance, while for SW, observed irradiance was 31 W m$^{-2}$ higher compared with ERA5.

To further understand the implications of these results, future studies should examine the relationships between the surface radiation fields and meteorological and aerosol parameters. This could help identify specific conditions under which the biases between observations and the ERA5 modelled cloud and radiation environment occur. Our analysis points to specific areas in the ERA5 representation for further investigation. These include the thermodynamics within the boundary layer and at heights below the LCL, the properties of clouds near the LCL in the sea ice zone, and more generally addressing the under-representation of clouds over the Southern Ocean.

**Supplementary Materials:** The following are available online at http://www.mdpi.com/2073-4433/11/9/949/s1, Figure S1: Frequency distributions of observations for sea ice and open ocean conditions as a function of latitude. Figure S2: Distributions of raw cloud transmittance versus solar zenith angle for sea ice and open ocean conditions. Figure S3: Histograms of differences between observations and ERA5 for SW and LW irradiance. Figure S4: Variation of the CRE difference between observations and ERA5 as a function of cloud fraction for open ocean and sea ice conditions. Figure S5: Variation of the CRE difference between observations and ERA5 as a function of lifting condensation level and ERA5 cloud base height for open ocean and sea ice conditions.

**Author Contributions:** Conceptualisation, S.P.A., A.R.K. W.J.R.F. and T.A.W.; methodology, H.W., A.R.K. and W.J.R.F.; software, H.W., W.J.R.W. and A.R.K.; validation, A.R.K. and W.J.R.F.; formal analysis, H.W. and A.R.K.; investigation, A.R.K. and H.W.; resources, T.A.W. and S.P.A.; data curation, W.J.R.F.; writing—original draft preparation, H.W.; writing—review and editing, A.R.K., W.J.R.F., S.P.A. and T.A.W.; visualisation, H.W. and A.R.K.; supervision, A.R.K., S.P.A. and W.J.R.F.; project administration, S.P.A.; funding acquisition, T.A.W. and S.P.A. All authors have read and agreed to the published version of the manuscript.

**Funding:** This research received no external funding.

**Acknowledgments:** This work contributes to Australian Antarctic Science (AAS-4292) and the Australian Antarctic Program Partnership and was assisted by the Antarctic Climate and Ecosystems Cooperative Research Centre and the Australian Government. We thank the Science Technical Support Team of the Australian Antarctic Division and the masters and crews of the RSV Aurora Australis and MS The World for their considerable efforts to make this work happen. In particular, we thank S. Whiteside, L. Symons, P. de Vries, C. Richards, C. Young and M. Cox for their assistance with instrumentation. We would also like to thank the resident/owners and resident Board of Directors of MS The World for their interest and cooperation in conducting scientific research aboard their vessel. Underway data for voyages of the RSV Aurora Australis are available through the Australian Antarctic Data Centre (https://data.aad.gov.au/).

**Conflicts of Interest:** The authors declare no conflict of interest.

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
