# Peer review of "Measurements of Cloud Radiative Effect across the Southern Ocean (43° S–79° S, 63° E–158° W)"

_atmosphere, doi:10.3390/atmos11090949_

Round 1

Reviewer 1 Report

Wang et al presents measurements of downwelling radiation over the Southern Ocean. From this they infer the cloud radiative affect, which they compare with ERA5 reanalysis data. They find that the ERA5 has a positive bias in the shortwave CRE and a negative bias in the longwave CRE. This dataset should be a useful resource to those studying the clouds in this region. The paper is well written with very few typos and the figures are a good standard.

I believe the paper should be published; however, some of the analysis could be extended, which would improve the utility of the paper. In particular, the reasons for the difference between the measurements and ERA5 can be explored further.

The paper finds that the modelled data underestimates cloud fraction. Can the differences between the measured and modelled CRE simply be explained by differences in cloud fraction? It should be possible to extract the cloud fraction from the model and do a comparison with the observations as a function of cloud fraction. Can you break down the CRE comparison by cloud fraction, cloud type/height/thickness and surface type.  I would like to see more information about the meteorological conditions during the observations and these should be used to inform the analysis. Were there particular periods/meteorological conditions when the agreement was better/worse?

I should make it clear that my area of expertise is in cloud microphysics not in radiation measurements. Therefore it is important that the other reviewers cover the description of how the radiation measurements were made, since it is outside my core area of expertise.

Introduction – You should discuss the uncertainty in modelling clouds and radiation in the Southern Ocean and how this paper can improve understanding of these. Clouds in this region have been poorly represented in numerical models e.g

Bodas-Salcedo, A., Williams, K. D., Ringer, M. A., Beau, I., Cole, J. N. S., Dufresne, J.-L., Koshiro, T., Stevens, B., Wang, Z., and Yokohata, T.: Origins of the Solar Radiation Biases over the Southern Ocean in CFMIP2 Models*, J. Climate, 27, 41–56, doi:10.1175/JCLI-D-13-00169.1, 2014.

Sect 2.7 – Could you show a histogram of (measured – modelled) SW and LW clear sky radiation. I suggest also giving the median and inter-quartile range of this in the text. This is a key piece of analysis that most of the conclusions of the paper are built upon.

Line 280 – could you provide a reference to the actual model the R subroutine uses?

Line 293 – ‘dis’ should be ‘does’

Figure 8 – For the error bars instead of showing the standard error of the mean could you show the inter – quartile range CRE, SCE and LCE for each cloud fraction bin. This is a more useful metric to show the variability in the population.

Reviewer 2 Report

In general, I think this paper is of good quality and worthy of publication. It provides a detailed analysis of radiative measurements over the Southern Ocean. This is a particularly important area of research due to the data sparsity and well established model biases in this region. However, I do think some changes are required for clarity, simplicity and conciseness.

Major comments:

  • I believe this paper could be improved by introducing some more references to previous work throughout the results section. This is done well in general but could add much to the discussion of figures 5, 6 and 12
  • Equations 1, 2 and 3 might be unnecessary due to the inclusion of equations 4, 5 and 6. I would recommend either removing one set of equations or changing the text in some way that would justify a separate set of equations.
  • Several figures could do with some refinement and tidying up. For example, the subplots in figures 4, 13 and 14 all need to be resized so that the subplots are the same size.
  • I think there are several figures that could potentially be removed to make the paper more succinct. Obviously the surrounding text would have to change to accommodate this.
    • Figure 9 seems relatively unimportant, I think it can be removed.
    • Figure 10 can potentially be removed as I don’t think it makes any points that aren’t clear from figure 11
    • While the results it contains are interesting, I am not sure if figure 14 adds much value to the paper in terms of conclusions. This particularly true of the results in panel a in which the cloud typing is interesting but seems out of scope considering the rest of the paper. If this figure remains I would recommend changing panel a

Minor Comments:

  • Line 17: reword “we summarize the radiation environment over the Southern Ocean within the region bounded by 42.8° S to 78.7° S and 62.6° E to 157.7° W during three Antarctic summers.” into something that follows better from the first part of the sentence. This might be best done by splitting the sentence in half
  • Line 60: The sentence that ends on this line could use a citation for completion.
  • Line 86: Change the sentence that starts on this line to something like “This higher occurrence rate of SLW clouds is likely due to the lower aerosol loading over the Southern Ocean.”
  • Line 90: Change the sentence starting on this line to be more clear. Possibly by deleting “which can provide more abundant data with higher accuracy during the voyage,”
  • Line 263: Reword the sentence starting on this line to be more clear. I think something like “The ratio of observed SW to clear sky conditions at the same solar zenith angle, known as the raw cloud transmittance (TRC), is also calculated using the voyage data.”. Depending on the exact wording used, equation 7 could be removed
  • Line 292: Change “This gave us confidence that” to “This demonstrated that”
  • Line 293: There is a typo with “dis”. I think it is meant to be does.
  • Line 302: The sentence that starts on this line would be much clearer if it was split into two sentences
  • Line 322: Is including the standard errors valuable here? This section might work better without it
  • Line 353: This paragraph could use an extra sentence offering some explanation for the differences identified in the last sentence
  • Line 356: The sentence beginning on  this line should be reworded for clarity. Maybe “The highest cloud fraction systems (cloud fraction over 90%) show a strong latitudinal dependence. These clouds are most common around 60 degrees and are significantly rarer poleward as clear sky conditions begin to dominate.”.
  • Line 383: The word “cover” appears incorrectly italicized
  • Line 403: I think that sentence beginning on this line doesn't make sense and should be rewritten 
  • Line 442: I think that sentence beginning on this line doesn't make sense and should be rewritten 
  • Line 470: “Further” should be changed to “Furthermore”
  • Line 497: Change “lower limit of values” to the “lower limit of CRE”
  • Line 502: It is unclear exactly what you mean by “shows more clearly the frequency of occurrence of observations”. Please change this.
  • Line 503: The sentence starting on this line should specifically mention that the ERA data is shown as red dots
  • Line 505: Reword the sentence that starts on this line for clarity. 
  • Line 550: Consider splitting this sentence in two as it is overly long 
  • Line 583: This sentence seems unclear and should be rewritten more concisely
